# Dynamically Controlling Offloading Thresholds in Fog Systems

**DOI:** 10.3390/s21072512

**Published:** 2021-04-03

**Authors:** Faten Alenizi, Omer Rana

**Affiliations:** School of Computer Science and Informatics, Cardiff University, Cardiff CF24 3AA, UK; RanaOF@cardiff.ac.uk

**Keywords:** fog computing, computational offloading, dynamic offloading threshold, resource management, minimizing delay, minimizing energy consumption, maximizing throughputs

## Abstract

Fog computing is a potential solution to overcome the shortcomings of cloud-based processing of IoT tasks. These drawbacks can include high latency, location awareness, and security—attributed to the distance between IoT devices and cloud-hosted servers. Although fog computing has evolved as a solution to address these challenges, it is known for having limited resources that need to be effectively utilized, or its advantages could be lost. Computational offloading and resource management are critical to be able to benefit from fog computing systems. We introduce a dynamic, online, offloading scheme that involves the execution of delay-sensitive tasks. This paper proposes an architecture of a fog node able to adjust its offloading threshold dynamically (i.e., the criteria by which a fog node decides whether tasks should be offloaded rather than executed locally) using two algorithms: dynamic task scheduling (DTS) and dynamic energy control (DEC). These algorithms seek to minimize overall delay, maximize throughput, and minimize energy consumption at the fog layer. Compared to other benchmarks, our approach could reduce latency by up to 95%, improve throughput by 71%, and reduce energy consumption by up to 67% in fog nodes.

## 1. Introduction

The number of IoT devices and their generated tasks are constantly growing, imposing a burden on cloud infrastructure, in particular if processing of these tasks must take place within Quality of Services (QoS) constraints [1,2]. The processing of these tasks in the cloud can trigger systems to suffer high communication latency, security issues, and network congestion [3]. This is due to the distance between IoT devices and cloud-hosted servers [4,5]. Fog computing has emerged to address limitation of processing IoT tasks at the cloud and ensure the processing of these tasks takes place within pre-defined time periods [6]. Fog computing is an intermediate layer situated between cloud and IoT devices that brings location awareness, low latency, and wide-spread geographical distribution for IoT devices [7,8]. It consists of limited-resource devices called fog nodes, providing storage, processing, and networking resources close to IoT devices where tasks are produced [8,9]. Fog computing was introduced by Cisco in 2012 [4,10]. With limited-resource devices used in fog systems, poor utilization of these resources would limit their benefit. 

Computational offloading enables workload/computational tasks to be shared between IoT devices, fog nodes, and cloud servers [11,12,13,14]. When computational offloading occurs between fog nodes, this is called “fog cooperation” [15], in which overloaded fog nodes send part of their workload to other underloaded fog nodes to meet their QoS requirements [16,17]. Resource management can involve multiple factors, saving energy consumption in the fog environment is one of these factors, and is considered in this work. Integrating computational offloading and resource management is essential to effectively utilize fog resources [1].

In online dynamic fog systems, where uncertainties are arising due to multiple factors, with no prior awareness of task arrival rate, the number of connected IoT devices, and computational capacity of fog nodes, addressing computational offloading and resource management is challenging to obtain optimum outcomes [1]. Computational offloading has mostly been explored in offline fog systems, where all system data are known beforehand, and limited work has been carried out in online dynamic fog systems. There is also limited work on understanding the impact of dynamically changing the offloading threshold, which is a factor that determines when a fog node begins sharing its workload with other neighboring fog nodes within its proximity.

### 1.1. Contributions

Specifically, this work provides the following contributions:We propose a fog node architecture that dynamically decides whether to process the received tasks locally or offloads them to other neighbors. This is based on a dynamic threshold that considers the queuing delay of the primary fog node and the availability (i.e., the queuing delay) of its neighbors.Computational offloading and the associated computational resource management was investigated using an online dynamic system with the aim to solve the multi-objective problem that aims to minimize delay, minimize energy consumption, and maximize throughput.We conducted extensive experiments to evaluate the performance of our proposed scheme and compare our proposed algorithm to various benchmarks.This paper extends our previous work [1] by introducing a dynamic offloading threshold, made use of in an online model for evaluating service delay.

### 1.2. Paper Organization

The remainder of this paper is organized as followed. Related work is provided in Section 2, followed by the system model and associated constraints in Section 3. In Section 4, we decompose the multi-objective problem into two sub-problems: delay minimization and energy saving, followed by a description of our solution in Section 5. In Section 6, we compare the performance of our proposed scheme against other benchmarks, followed by conclusions in Section 7.

## 2. Related Work

This section is divided into three main parts. The first focuses on computational offloading between entities within a specific system; the second addresses the impact of dynamically managing servers to enhance power efficiency. Finally, a comparison of state-of-the-art of related approaches in fog computing is provided and summarized in Table 1.

### 2.1. Computational Offloading

Computational offloading can be implemented offline or online. In offline implementation, all system information needed to make the offloading decision is previously known and is based on historical or predictive knowledge, such as the computational capabilities of fog nodes, the total number of IoT devices, and their total workload (number of requests). This is applied during the system design stage. In online deployment, the computational offloading decision takes place at run-time and considers the current system status and process characteristics, such as the current waiting time and the current available computational resources, without prior knowledge of system inputs considered in the offline deployment. Several studies investigate computational offloading in offline deployment [8,11,12,13,14,18,19,20,21,22,23]. In [19], Wang et al. investigated the optimized offloading problem to minimize task completion time given tolerable delay and energy constraints. The optimization problem was formulated as a mixed integer nonlinear programming problem that jointly optimizes the local computation capability for IoT devices, the computing resource allocation of fog nodes and the offloading decision. Wang et al. [19] decomposed it into two independent sub problems to find the optimal amount of workload that should be processed locally at IoT devices and at fog nodes. A hybrid genetic-simulated annealing algorithm was developed to optimize the offloading decision. Tang et al. [18] aimed to increase the total number of executed tasks on IoT devices and fog nodes under deadline and energy constraints. The authors in [18] considered this as a decentralized, partially observable offloading optimization problem in which end users are partially aware of their local system status, including the current number of remaining tasks, the current battery power, and the nearest available fog node. Such parameters are used to assess if tasks should be processed locally or offloaded to the nearest fog node. Their approach enables IoT devices to make an appropriate decision based on its locally observed system.

Liu et al. [11] addressed a multi-objective optimization offloading problem in a fog environment with the aim of minimizing execution delay, energy consumed at mobile devices, and offloading payment cost for using fog/cloud resources. The multi-objective problem was formulated into a single problem using scalarization method [11]. The proposed solution found the optimal offloading probability that accomplishes the stated objectives. Mukherjee et al. [20] designed an offloading technique focusing on jointly optimizing the computing and communication resources at fog systems to reduce end-to-end latency. Their technique considers the trade-off between transmission delay and task execution delay when making the offloading decision, in which a fog node can seek additional computational resources from either one of its neighbors, or the cloud data center, to reduce task execution delay at the expense of the transmission delay. The optimization problem was transformed into convex quadratically constraint quadratic programming and solved using CVX toolbar, which is a MATLAB-based modelling system for convex optimization. Their simulation results demonstrated that their proposed solution offers minimal end-to-end latency in comparison to executing all tasks at end-user devices and executing all tasks at the primary fog nodes.

Zhu et al. [13] proposed a task offloading policy based on execution time and energy consumption. This approach helps mobile devices to make an appropriate decision on whether to process their tasks locally or offload them to a fog node, or the cloud. During the decision-making procedure, mobile devices calculate both the execution time and the energy consumed when executing the task on the local device and compare this with the execution time and the energy consumed when offloading and receiving the processed task on a fog node; the energy consumed when executing the tasks on fog nodes are not considered. Based on this comparison, the IoT device makes a decision with the least cost (execution time plus energy consumption). Comparing their scheme to Random, no offloading, and only offloading when considering only execution time, their simulation results showed an optimization of the execution time of tasks and energy consumption of mobile devices. Mukherjee et al. [21] formulated the offloading problem as an optimization problem with the goal to minimize the total system cost, which is the sum of the total delay of end-users’ tasks and the total energy consumed at end-users’ devices due to local processing of tasks and uploading tasks to the fog environment for processing. Under delay and energy constraint, the optimization problem was transformed into a quadratically constraint quadratic programming problem and solved by semidefinite relaxation method. Within a heterogeneous environment where fog nodes have different computational resources, the proposed solution enables the optimal amount of workload to be identified that should be processed at end-user devices, primary and neighboring fog nodes, and cloud servers. The decision on when to offload depends entirely on the availability of computational resources. The authors stated that having higher computational resources at fog nodes helps to reduce the system cost. Increasing number of end-users leads to greater congestion at fog nodes, leading to fog nodes preferring to send their workload to the cloud server for processing rather than other neighboring fog nodes.

Chen and Hao [14] studied offloading problem in dense software-defined networks, formulating this as a mixed-integer nonlinear problem that is decomposed into: (i) deciding whether the task is processed locally at the end-user device or offloaded to the edge device; (ii) determining the computational resources that are dedicated to each task. Chen and Hao [14] developed an efficient software-defined task offloading scheme to solve these sub-problems. The results of their proposed scheme demonstrated the superiority of their approach at decreasing end user device energy consumption and overall task execution latency. In IoT-Fog-Cloud architecture, Sun et al. [22] presented the “ETCORA” algorithm, which consists of two parts. The first part aims to find the optimal offloading decision based on minimizing time and energy, and the second part optimizes resource allocation in terms of transmission power allocation. Their proposed solution helps to minimize energy consumption and completion time of tasks compared to other schemes. Zhao et al. [12] investigated the computational offloading problem in the context of radio access networks to reduce the weighted sum of total offloading latency plus total energy consumption. To improve the offloading decision and enhance the allocation of computation and radio resources, the authors formulated the problem as a non-linear, non-convex joint optimization problem. Their proposed solution was more effective than mobile cloud computing (MCC), which processes all end-user tasks on a cloud server, and mobile edge computing (MEC), which processes all end-user tasks in the edge computing system. The reason their approach was more effective compared to MCC and MEC is that it made use of a combination of available resources at the cloud and fog nodes, compared to *cloud only* as in MCC, and edge only as in MEC.

The hybrid-computational offloading optimization problem was investigated by Meng et al. [23], where two types of models were considered; namely cloud computational offloading and fog computational offloading. The authors aimed to minimize the consumption of energy caused by transmitting and processing tasks at mobile terminals, fog, and cloud servers under deadline constraints. Meng et al. [23] introduced a new concept called *computation energy efficiency* that is defined as “the number of computation tasks that are offloaded by consuming a unit of energy”, to solve the optimization problem. Based on the proposed solution that considers offloading tasks to fog and cloud servers for execution, simulation results show the effectiveness of the solution compared to only offloading tasks to either cloud only or fog only resources. Xiao and Krunz [8] proposed a workload scheduling method that ensures user response time is minimized under available power constraints. In their study, the energy spent while processing tasks was ignored and only the energy consumed for offloading each unit of received workload was considered. Cooperation between fog nodes to offload workload by an agreement between neighboring nodes, the workload arrival rates, and the workload processing capabilities determines the amount of offloading carried out. Their experimental results indicated that the average response time decreased due to allowing cooperation between fog nodes. Additionally, a crucial trade-off between the fog node’s power efficiency and the average response time was observed. Xiao and Krunz [8] proposed that the response time of end-user tasks should be set to its highest tolerable point to optimize energy consumption at fog computing systems. This enables most of the tasks to be processed at end-user devices, avoiding any offloading.

Regarding online deployment of computational offloading, few studies have addressed this, such as [15,16,17,24,25,26]. Yousefpour et al. [16] suggested a delay-minimization approach to reduce overall service delay. In their approach, the estimated queueing delay, which is utilized as the offloading threshold, determines whether a fog node processes its incoming task(s), or offloads these to one of its neighbors or the cloud server. If the offloading threshold has been reached, then the best neighboring fog node in its domain is selected to offload its upcoming tasks. The best neighboring fog node is chosen based on having the minimum total of propagation delay and queuing delay. Compared to other models, their results achieved the minimum average service delay. Yin et al. [24] determined where to process end user tasks into task scheduling and resource allocation problems, where tasks are either processed locally at end-user devices or offloaded to fog nodes or cloud servers. In an intelligent manufacturing environment, the authors introduced fog computing and utilized the concept of the container within the fog system, intending to reduce overall delay and optimize the number of concurrent tasks for the fog node. In their online model, generated tasks by end-users are transmitted to the request evaluator, which is located at a fog node that decides whether to accept or reject the task based on its deadline requirement. If the task is accepted, then the task is transmitted to the task scheduler, which determines whether the task is processed at fog nodes or cloud servers based on the available resources and the execution time of this task, which involves computation and transmission time. Finally, the resource manager is responsible for reallocating the required resources to process the task at fog nodes. Experimental results showed the effectiveness of their approach compared to other benchmarks.

Al-Khafajiy et al. [15] proposed an offloading mechanism that allows fog-to-fog collaboration in heterogeneous fog systems, intending to minimize overall service latency. Their mechanism utilizes a FRAMES load balancing scheme that aims to detect congested fog devices, determine the amount of workload located at fog devices’ queues that require offloading, based on their deadline requirement, and finally select the best fog node that provides the minimal service latency for the selected workload. They evaluated their proposed mechanism using a simulation. Their numerical results indicated the effectiveness of their proposed model in terms of minimizing overall latency in comparison with different algorithms. In a fog-cloud computing system, Gao et al. [17] investigated the issue of dynamic computational offloading and resource allocation. In order to reduce energy consumption and delay while having a stable queueing status, the authors formulated the problem as a stochastic network optimization problem. They provided a predictive approach to computational offloading and resource allocation that depended on the trade-off between delay and energy use. Their approach implied that a delay reduction can be induced by increasing the allocation of computational resources at fog nodes; however, because of the processing of more tasks, energy consumption increases, and vice versa. Compared to other systems, the authors showed the importance of their method. Mukherjee et al. [25] developed a scheduling strategy that managed to fulfil the deadline constraint of end-user tasks, taking into account computational resources. The deadline constraint of a given task and the availability of a neighbor, in their scheduling policy, help to decide on whether to place a given task in the fog node queue, e.g., in its high-priority queue or low-priority queue, or offload it to one of its neighboring fog nodes. Their findings illustrated the efficacy of their suggested strategy as opposed to the no offloading and random schemes. Table 1 presents a summary of relevant articles concerning computational offloading at fog computing systems and the forms in which these systems execute.

### 2.2. Dynamic Server Energy Management

Dynamic Server Energy Management has been used in the wireless local area network and the cloud, and it has proven to be efficient in terms of improving power quality. Although up to the time of our study, this has not yet been implemented in the fog area. In WLANs, the energy efficiency was enhanced by placing access points (APs) in sleep mode or turning them off. In [27], Marsan and Meo observed that in a community of APs, getting one AP in each community to control the system and service the incoming clients when all others are turned off will minimize energy consumption by up to 40 percent. Furthermore, an additional 60% of consumed energy can be saved if all APs are turned off, especially during idle periods, e.g., at night. Li et al. [28] suggested an energy-saving method for state transformations in which APs are not only turned on and off based on consumer requirements, but there is also an intermediary stage that aims to reduce the frequency of switching. The authors stated that increasing the switching frequency will shorten AP’s service life. In addition to that, the intermediary stage will help to avoid latency and energy overhead caused by switching on APs.

It was suggested that servers could be periodically switched off [29,30] or placed into sleep mode [31,32,33] in cloud computing systems to conserve energy resources. In [29,30,31,32,33], the authors examined the issue of the placement of virtual machines (VMs) to save resources concerning energy and yet retain QoS. When underutilized data centers are detected, all VMs are migrated to other active data centers, and these underutilized data centers are placed in sleep mode according to [31,32,33] or shutdown as per [29,30]. This is intended to reduce the consumption of energy at cloud computing systems and is called ‘VM consolidation’. Numerous VM migration approaches were suggested to assess which virtual machines can be migrated from overloaded data centers. Moreover, in order to satisfy the QoS specifications of the system, a switched-off data center may also be activated to handle the migrated VMs. According to Mahadevamangalam [31], the energy demand for an idle data center is ~70% percent of the energy generated by a fully utilized data center. Thus, by switching off idle-mode data centers, up to 70% of the energy consumed can be saved in the cloud system.

### 2.3. Comparison of the State-of-the-Art

Table 1 provides a summary of related work in computational offloading in fog computing systems, highlighting the architecture model, e.g., IoT-Fog means that end-user tasks are processed locally at IoT devices or offloaded and processed at fog nodes, use of fog cooperation, communication, the stated objectives of the work, and evaluation tools arranged by offline or on-line offloading decisions. Offline deployment helps to predict the best output for the system at its design stage; and online deployment mimics various scenarios in real-world environments, involving uncertainty and unpredictable events, and helps the system to produce a better outcome. However, most of the literature is focused on offline deployment. Additionally, the problem of computational offloading is usually investigated with the aim of minimizing an overall delay in the system; managing system resources is sometimes included, especially minimizing energy consumption of IoT/end user devices.

Managing resources in the system is much easier within offline deployment than online deployment, especially when all the system data is known in advance. In offline deployment, most attention has been given to addressing the energy consumed at IoT devices compared to fog devices. Additionally, when considering energy consumed at fog nodes, often the trade-off between delay and energy has been investigated.

In this work, we consider online deployment of computational offloading and the potential for minimizing energy consumption at fog nodes (compared to energy consumption of networks or cloud servers). Computational offloading and resource management at fog environments has received limited attention so far. When considering computational offloading, existing efforts utilize a fixed threshold that determines when to start offloading; in the current work, a dynamic threshold is investigated to address its impact on the system.

## 3. System Modelling and Constraints

Based on the model in [1], we describe an extended fog node architecture in Section 3.1.3.

### 3.1. System Model

The proposed model consists of one cloud server, ‘N’ fog nodes that are located at roadside units (RSU), a fog controller, and M vehicle nodes. Each vehicle node connects to the associated fog node through a wireless local area network, and the connection to the remote cloud server is via a wide area network. A single task contains the following data, T = {Type, S_m_, D_m_, Task^CPU^, Task^Network^}, where Type is the category of task being considered (e.g., urgent or non-urgent); S_m_, D_m_ respectively represent the source application module (from where the task is emitted) and the destination application module (where the task is heading); Task^CPU^ indicates the computational complexity of the tasks, captured in number of instructions (Million Instructions Per Second (MIPS)); Task^Network^ represents the size of the encapsulated data in the task that needs to be transmitted across the network. In iFogSim, the simulator used to model the system, tasks are represented as tuples. A network diagram is presented in Section 3.1.1 and the associated application module is described in Section 3.1.2.

#### 3.1.1. Network Diagram

Figure 1 shows an illustration of the fog computing architecture, which comprises of three layers:**The IoT devices layer:** This layer is composed of mobile vehicles—represented as vehicle nodes, containing an actuator and a collection of sensors. Each sensor produces a task, labelling it as “non-urgent” or “urgent”. Non-urgent tasks include data such as current position, speed, and path. Urgent tasks require a quicker response and can have stringent Quality of Service (QoS) requirements. This task may contain a video stream around a moving vehicle, requiring short latency or processing. This is necessary, for instance, in self-driving vehicles.**Fog computing layer:** This layer is comprised of a series of fog nodes and a fog controller. Fog nodes are located in RSU that are installed alongside a road. If fog nodes are situated in communication proximity of each other, they can interact and share data with each other [34]. Hence, fog nodes form an ad hoc network to exchange and share data. All fog nodes are linked to the fog controller, which is responsible for managing fog resources and controlling fog nodes. Fog nodes process two different types of tasks, urgent tasks are given priority and their processing results are sent back to the vehicle. For non-urgent tasks, fog nodes process these tasks and transfer the findings to the cloud for further analysis and storage, e.g., for retrieval by traffic management organizations.**Cloud computing layer:** This layer is composed of a set of cloud servers, hosted within one or more data centers. This layer is able to aggregate traffic information across a geographical area over time.

#### 3.1.2. Application Module Description

Three modules: road control, global road monitor, and process priority tasks are part of the application model. The first two modules focus on traffic light management, while the last module is responsible for processing urgent tasks. The operations carried out by these modules are outlined below.
**Road Monitor:** This module is placed at fog nodes. When a vehicle comes into communication proximity of a fog node, a sensor immediately sends data to the connected fog node for analysis. This data contains the current position, the speed of the vehicle, weather, and road conditions. After processing these data by the specified module, the results are transmitted to a cloud server for further processing.**Global Monitor:** this module is placed at a cloud data center, receiving data that has already been processed by the road monitor module.**Process Priority task:** this module is placed at fog nodes and is responsible for processing priority requests from a user. The results are then sent back to the user. An application in iFogSim is specified as a directed acyclic graph (DAG) = (M, E), where M represents the deployed application modules M = {m_1_, m_2_, m_3_, …, m_n_}, e.g., process priority task, road monitor, and global road monitor modules. ‘E’ represents a set of edges describing data dependencies between application modules, as illustrated in Figure 2.

#### 3.1.3. Fog Node Architecture

The proposed fog node architecture consists of a task scheduler, best neighbor selector, and threshold monitor (see Figure 3). Task scheduler receives tasks generated from IoT devices within the proximity of the primary fog node and from other neighboring fog nodes. If a fog node receives a task that is already offloaded from another neighbor, task scheduler immediately inserts this task in the processing queue. If the task is generated from other IoT devices, then task scheduler will check the offloading threshold and compare this to the queuing delay at the current node. If the queueing delay reaches the offloading threshold, then task scheduler sends this task to the best neighbor selection, which in turn decides the best neighbor node to offload this task to. The selection of the best neighbor is described in more detail in Section 3.2.2. Threshold monitor is responsible for dynamically increasing and decreasing the offloading threshold for both the primary fog node and all its neighbors, based on the workload and the availability of other neighbors: this is done from the perspective of the primary fog node. On the one hand, it is assumed that fog nodes are cooperative and accept tasks coming from their neighbor nodes, even if this exceeds their threshold. On the other hand, each neighbor has its own threshold monitor, and the primary fog node and all its neighbors may not have the same threshold value. In Table 2, we can see that primary fog node A set its threshold to 9 ms for itself and all its neighbors. At the same time, primary fog node B in Table 3 set its threshold to 6 ms, even for its neighbor fog node A; therefore, it can be seen that fog node A is congested and will not be selected as the best neighbor for fog node B. Determining when to increase and decrease the offloading threshold is described in Section 5.

### 3.2. Types of Connections and Constraints

This section explains the relations between a vehicle and a fog node, between fog nodes, and between fog nodes and cloud servers. Additionally, we also specify the restrictions that render these relations appropriate.

#### 3.2.1. Connection between Vehicles and Fog Nodes

The interaction between a vehicle and a fog node is controlled by communication and processing restrictions.

Communication Constraints

Each vehicle connects to a fog node if it is located within the communications coverage radius of that fog node, as specified in Constraint (1).
(1)$$D_{v,\;f}\leq \;max\;Coverage_{f};\;\forall \;v\in V,\;\forall \;f\in \;FN$$
where *V* represents all vehicles, *v* is a single vehicle, *FN* represents all fog nodes, and *f* is a single fog node. *D*_*v*,*f*_ is the distance between a vehicle *v* and a fog node *f*, and calculated as:(2)$$D_{v,\;f}=\;\sqrt{{\left(X_{v}-\;X_{f}\right)}^{2}+{\left(Y_{v}-\;Y_{f}\right)}^{2}};\;\;\;\;\forall \;v\in V,\;\forall \;f\in \;FN$$
where (*X_v_*, *Y_v_*) and (*X_f_*, *Y_f_*) are the location of the coordinates of a vehicle *v* and a fog node *f*, respectively. When a vehicle is within a range of several fog nodes, it will connect to the nearest fog node. This is to decrease delay, as the propagation delay relies on the distance between the two connected nodes, propagation delay (*PD*) is calculated as
(3)$$PD=\frac{D_{v,\;f}}{PS}$$

Signal propagation (*PS*) speed is assumed to be equivalent to the speed of light [35], i.e., *c* = 3 × 10^8^.

Processing Constraints

To enable placement of application modules, fog nodes should have enough resources to meet the demands of these application modules.
(4)$$\overset{M}{\underset{i=0}{\sum }}Required_{Capacity}m_{i}\leq \overset{FN}{\underset{j=0}{\sum }}Available_{Capacity}f_{j};\;\forall \;m_{i}\in M,\;\forall \;f_{j}\in \mathrm{FN}$$

The required capacity of an application module and fog node is therefore captured using *CPU*, RAM, and bandwidth. Constraint (4) indicates that the overall needed capability of all application modules should not surpass the available capacity of the fog node on which they are installed. In the iFogSim simulator, if there is no available capacity at fog nodes, the application will be placed at cloud servers. The required *CPU* capacity to place an application module is calculated as followed:(5)CPU=NV×(Rate×TaskCPU)
where *NV* is the number of vehicles attached to the fog node, *TaskCPU* is the number of instructions contained in each task, specified in Million Instructions Per Second (MIPS). Rate is calculated as:(6)$$Rate=\frac{1}{Transmission\;Time\;in\;ms}$$

In iFogSim, application module placement takes place at the design stage. Increasing the number of connected vehicles at a fog node will increase the required *CPU* requirement, to execute the required application modules. If the fog node does not have enough *CPU* capacity, these applications will be placed in the cloud. In this case the number of connected vehicles for each fog node is limited, as specified in Constraint (7).
(7)$$\overset{V}{\underset{i=0}{\sum }}v_{i}f_{j}\leq \;MAX_{vehicles};\;\forall \;v_{i}\in V,\;\forall \;f_{j}\in \;FN$$

#### 3.2.2. Connection between Fog Nodes

In this section, we describe the waiting queue for fog nodes, based on which an offloading decision is determined.

Fog Node Waiting Queue.

All fog nodes contain a queue for arriving tasks, served on a sequential First In First Out (FIFO) basis. A queueing delay triggers the decision to begin offloading tasks from the arrival queue to neighboring fog nodes [16]. To begin offloading tasks, the queue waiting time should exceed the predetermined offloading threshold.
(8)$$T^{Queue}\;>{Offloading}_{threshold}.$$

*T^Queue^* is calculated as
(9)$$T^{Queue}=\;\sum T_{i}\times \;T_{i}^{process}+\;\sum T_{z}\times \;T_{z}^{process};\;\forall \;i,z\;\;\in T$$
where *T_i_* and *T_z_* are the total number of tasks of the type *i* and *z*, urgent or non-urgent, respectively. *T* is all tasks and is the expected execution time of a specific task and calculated as
(10)$$T^{process}=\;\frac{TaskCPU}{F\_MIPS\times N\;of\;PS}$$
where *F_MIPS* is the total computational capacity (measured in *MIPS*) available at a fog node, and ‘*N* of *PS’* is the total number of processing units at that fog node.

Coverage Method

Fog nodes can have overlap in their coverage area [35], as illustrated in Figure 4.

Selecting the Best Neighboring Fog Node

Fog nodes form an ad hoc network between them to share and exchange data such as their queueing delay. Following [16], the best neighboring fog node is selected based on propagation delay plus queueing delay. The selection of the *best* neighboring fog node begins if the offloading threshold of a fog node is reached, determined by the waiting queue time. A fog node can communicate with neighbors in its coverage area, as specified in Constraint (11)
(11)$$d_{ij}\leq \;Coverage_{radius};\;\forall \;i,\;j\;\in \;FN$$
where *d_ij_* represents the distance between fog nodes *i* and *j*. In Figure 4, the neighboring fog nodes for FOG 1 are FOG 2 and FOG 3. Additionally, the neighboring fog nodes for FOG 3 are FOG 1, FOG 4, and FOG 5. The criterion for choosing the best neighboring fog node is based upon the coverage radius of the primary fog node, and the sum of queueing and propagation delay to the neighbor, where PD is calculated in (3) above.
(12)$$Min\;\;\sum T^{Queue}+PD$$

#### 3.2.3. Between Fog Nodes and the Cloud

We primarily focused on sharing workload with other neighboring fog nodes in preference to using cloud servers. This is attributed to the availability of other neighboring fog nodes as they overlap and to get maximum utilization of the available resources in the fog system. Task exchange between fog nodes and the cloud is only considered if the primary fog node and all its neighbors are congested, e.g., all their offloading thresholds reach the maximum threshold. We assume that cloud servers are much more efficient, and their queueing latency is ignored—i.e., tasks are processed immediately upon arrival at a cloud server [36,37,38]. The maximum offloading threshold is calculated as followed:(13)$$\mathrm{Maximum}\;\mathrm{threshold}=\;2\times \left({Transmission}_{cloud}^{delay}\right)$$

## 4. Problem Formulation

The multi-objective problem of minimizing delay, maximizing throughput, and minimizing energy consumption is decomposed into two sub-problems [1]: (i) delay minimization and throughput maximization, and (ii) energy saving.

### 4.1. Delay Minimization and Throughput Maximization

The response time is the time required for sending the workload from a vehicle to the connected fog node and getting the results back. It consists of the transmission delay, propagation delay, queuing delay, and processing delay. If task processing takes place at the primary fog node, then the service latency is calculated as:(14)$$T\;=\;T^{sTv}\;+2\;\times \;(T_{vTf}^{Transmision}\;+PD_{vTf})+T^{Queue}\;+\;T^{procss}+\;T^{vTa}$$
where $T^{sTv}$. and $T^{vTa}$ are the latency time between a vehicle and its sensor, and between the vehicle and its actuator, respectively. $T_{vTf}^{Transmision}$ is transmission delay between the vehicle and its primary fog node. It is based on the network length of the task (i.e., its data size) and the available bandwidth, and is calculated as:(15)$$T^{Transmision}=\;\frac{Network\;Length\;of\;Task}{Bandwidth}$$

If a neighboring fog node is used to carry out processing of the received task, then latency is calculated as:(16)$$T=\;T^{sTv}\;+2\times \;T_{vTf}^{Transmsiion}+\;PD_{vTf}\;+2\times \;T_{fTf}^{Transmission}\;PD_{fTf})+T^{Queue}+T^{Process}\;T^{vTa}$$

If the cloud is incorporated in the processing of the task, then the latency is calculated as:(17)$$T=\;T^{sTv}\;+2\times \;T_{vTf}^{Transmsiion}+\;PD_{vTf}\;+2\times \;T_{fTc}^{Transmission}\;+\;T^{Process}\;T^{vTa}$$

Throughput is measured as the percentage of the processed tasks as the following.
(18)$$Throughputs=\;\frac{total\;number\;of\;processed\;tasks\;in\;the\;system}{total\;number\;of\;genertaed\;tasks\;in\;the\;system}\times 100$$

### 4.2. Energy Saving

Minimizing the power consumption of fog nodes brings many advantages, including but not limited to decreasing the overall cost of electricity and reducing the environmental impact. Two power modes are presented for each fog node: idle and busy. In the idle mode, the fog node is not performing any processing, but the power is ON, and in the busy mode the fog node is processing tasks and power is ON. The energy consumed is determined by how much power the fog node consumes when processing workload and when the fog node is idle. The total energy consumption in iFogSim is calculated [39] as:(19)E=PR+(TN−LUT)×LUP
where *PR* is the previously calculated total energy consumed at this fog node, *TN* is the time now, which is the time that the updateEnergyConsumption() is called when utilizing this fog node, updateEnergyConsumption() is a method located at Fog Device class in iFogSim, LUT is the last time this fog node has been utilized, and finally LUP is the last used power status (for either idle or busy period). The problem of minimizing delay and energy is formulated as followed:Min ∑T  & ∑E, s.t. (1), (7) and (4) hold
(20)$$T^{Queue\;}\;\leq Offloading_{threshold}$$
(21)$$P_{F}\;+\;P_{N}=1,\;P{\;}_{F}\;\&\;P{\;}_{N}\;=\;\left\{0,\;1\right\}$$

Equation (1) ensures the connection between a fog node and a vehicle that is located within its coverage range. Equation (7) guarantees the number of vehicles connected to a fog node does not exceed the threshold number. Constraint (4) ensures the placement of the required application modules at fog nodes. Equation (20) ensures the stability of fog node queues so that to process incoming tasks, the waiting queue time should not exceed its threshold. In constraint (21), $P_{F}$ (Primary Fog), and $P_{N}$ (Primary Neighbor), i.e., $P_{F}$ = 1 and $P_{N}$ = 0 if the task is processed on the node where it is generated.

## 5. Proposed Algorithms

Two algorithms are proposed [1] called dynamic task allocation (DTA) and dynamic resource saving (DRS)—which need to be combined. Previously, these algorithms were applied to a static offloading threshold [1]. In the current work, we proposed a dynamic offloading threshold, in which the offloading threshold is adapted based on the workload and the availability of neighboring fog nodes, as described in Section 5.1.

### 5.1. Dynamic Offloading Threshold

The dynamic threshold is managed by the threshold monitor, which adjusts its value periodically according to the received workload and the availability of other neighbors, as described in Algorithm 1. The first part of the algorithm (Procedure 1) determines whether to increase the offloading threshold of the primary node and its neighbors. This runs each time a new task arrives at the primary fog node. It starts by checking if the current threshold exceeds the maximum offloading threshold calculated in Equation (13), if this occurs then the best decision for the arrival task is to be migrated to the cloud for processing. Otherwise, it checks whether the queuing delay of the primary fog node has reached its offloading threshold, i.e., to decide whether to process the task locally and add it to its queue or select the best neighbor with the least queueing delay as per lines 4–16. The current threshold is then updated using Equation (23) and Procedure 2 is called.

The second part of the algorithm (Procedure 2) determines whether the threshold should be decreased. This runs each time a new task is received, and when the fog node finishes the execution of a task. It starts by checking if the current threshold of the primary fog node is larger than a threshold, as per line 25. If this occurs, then the average queueing delay for all the neighbors is calculated as in (22) and the current threshold is updated, as per lines 26–27. The computational complexity of the proposed algorithm is O(n). Parameters used in this algorithm are fin Table 4.
(22)$$VQ=\;\frac{\sum_{s=0}^{Ns}Qs}{Ns}$$
(23)$$\delta^{n+1}=\left\{\begin{array}{cc} \delta^{n}-p, & Q\geq \alpha ,\;VQ<x\\ \delta^{n}, & Q\geq \alpha ,\;VQ\geq x\\ \delta^{n}, & Q<\alpha \\ \delta^{n}+p, & Q\geq \delta^{n},\;VQ\;\geq \;\delta^{n} \end{array}\;\right.\forall \;x,\;\alpha ,\;p\;>0$$

   **Algorithm 1** Dynamic Offloading Threshold

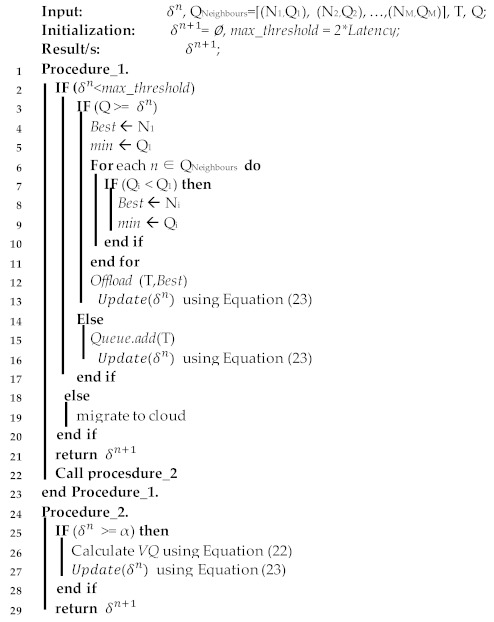



### 5.2. Dynamic Resource Saving

We used the algorithm from [1] to reduce energy consumption at fog nodes by dynamically switching ON/OFF fog nodes.

## 6. Experimental Results

In this work, iFogSim was used to simulate the environment. It was a toolkit developed by Gupta et al. [40], as an extension of the CloudSim simulator. It was a toolkit allowing the modelling and simulation of IoT and fog environments and can monitor various performance parameters, such as energy consumption, latency, response time, cost, etc. Simulation settings values were used as in [1]. The simulation was run with one cloud server, seven fog nodes, a fog controller, and a total of 50 vehicles. Each vehicle transmitted two different tasks from two sensors every 3ms. In iFogSim, the workload was represented as tuples, generated from vehicle nodes, and the following main classes were considered. FogDevice class was used to define the main characteristics of fog nodes and cloud servers, including RAM size, processor capacity in MIPS, uplink and downlink bandwidth, idle and busy power. Sensor class in the FogDevice class represented the attributes of a vehicle sensor, such as the vehicle node id to which the sensor is connected and the latency between them. Tuple class was used to represent computational tasks. The metrices used to measure the performance were:**Service Latency** is the average round trip time for all tasks processed in the fog environment. Two control loops were used in the simulation: **Control loop A:**
*Sensor -> Process Priority Tasks -> Actuator*. This control loop represented the path of priority requests. **Control loop B:**
*Sensor -> Road Monitor -> Global Road Monitor*. This control loop represented the path of non-priority requests.**Throughput**, which was measured as the percentage of processed tasks within a time window.**Total Energy Consumption** in fog environment caused by powering on fog nodes and processing tasks.

### 6.1. Performance Comparisons with Various Computation Offloading Schemes

To evaluate the effectiveness of our proposed algorithm, the comparisons with various computation offloading schemes were provided, where the number of vehicles was set to 50 and total number of fog nodes was set to 7. Although we implemented uncertainty within the system to mimic real world scenarios, we maintained the number of total generated tasks, the capacity of fog nodes, and the size of the generated tasks to be identical for fair comparison. In particular, the following four schemes were selected as benchmarks:

**Benchmark 1:***No Offloading Scheme (NO):* in this scheme, each primary fog node processes all the tasks without cooperation with other neighboring fog nodes.

**Benchmark 2:***Joint Task Offloading and Resource Allocation Scheme (JTORA)* [20]: in this scheme, if the primary fog node does not have enough computational resources that meet the delay requirement of a task, then the task will be offloaded to a neighboring fog node within the proximity of the primary fog node that has enough computational resources. Any underutilized neighbor is a candidate of processing the overload, ignoring the selection of the least utilized fog node. In this scheme, a static threshold is applied.

**Benchmark 3:***Workload Offloading Scheme (WO)* [26]. In their work, end users offload their computational tasks to a broker node that manages the system, the broker node will send tasks to a fog node closest to end users (primary fog node). If the primary fog node is congested (e.g., its queueing delay reaches 50 ms), then the broker node will offload the task to any underutilized neighboring fog node. In this scheme, a static threshold is determined.

**Benchmark 4:***Static Threshold 50ms Scheme (ST50)* [1]: where offloading threshold is set to 50 ms, upon which the primary fog node makes the decision on whether to process the task locally or offload it to the best neighboring fog node. The four benchmarks are compared to the proposed offloading policy called Dynamic Threshold (DT).

### 6.2. Impact of the Proposed Scheme and Different Offloading Schemes on Delay and Throughputs

In Figure 5, the impact of various offloading schemes on average latency is addressed. It can be seen in Figure 5 that delay was very high in the no offloading scheme; this was due to a long queueing delay as tasks were not shared by the primary node with other neighboring fog nodes, so they were waiting to be executed by the primary fog node. The impact of allowing cooperation between fog nodes in terms of sharing workload is shown, comparing other schemes to the no offloading scheme. Additionally, the impact of selecting which neighbor to share the workload with was very clear when comparing the WO, JTORA, ST50, and DT schemes. In the WO and JTORA schemes, when the primary fog node was congested (e.g., reaching its offloading threshold), it selected any underutilized neighbor to share the workload, rather than selecting the least utilized neighbor, as in ST50 and DT.

In addition, the delay was higher in the WO scheme compared to JTORA; this was due to a communication overhead caused by sending tasks to a broker node first, which in turn decides whether to process these tasks at the primary fog node or any underutilized neighbor. The least delay was achieved for both control loops when applying our proposed algorithm, DT, compared other benchmarks.

The impact of various offloading schemes on throughput is shown in Figure 6. It can be seen that the lowest percentage of processed task was when no offloading was applied; this was obvious as most of the tasks were waiting in the queue to be executed by the primary fog nodes. The impact of sharing workload with any underutilized neighbors was very clear in WO and JTORA schemes, resulting in processing almost 90.88 and 91.06% of tasks, respectively, compared to 94.56 and 95.67% in ST50 and DT schemes, respectively.

### 6.3. Impact of Increasing Number of Vehicles on Delay and Throughputs with Different Offloading Schemes

The impact of increasing the number of vehicles was to investigate how the delay was maintained as we increased the workload in the online system. In this experiment, the total number of fog nodes was set to seven and the number of vehicles ranged from 4 to 48. In Figure 7, we can observe that when the number of vehicles was small, between 4 to 12 vehicles, the DT, ST50 and JTORA schemes exhibited an identical pattern. This was because the generated workloads were small, resulting in the primary fog nodes processing most of these workloads themselves. When the number of vehicles increased, all three approaches, ST50, JTORA, and WO, showed a dramatic increase in delay compared to DT, which displayed a stable pattern with a slight increase in delay that increased as the number of vehicles increased.

The reason for the huge increase in delay for ST50, JTORA, and WO was that increasing the workload made the primary fog nodes almost reach their offloading threshold (e.g., 50 ms), but not always exceeding it, resulting in the primary nodes processing most of the workload with little help from neighboring nodes. The impact of selecting the best neighbor to share the workload becomes clear when the number of vehicles is high (i.e., 28 vehicles). The overall results show the effectiveness of the DT scheme even when increasing the number of vehicles.

The impact on throughput was also investigated while increasing the number of vehicles. When there was a small number of vehicles, ranging from 4 to 12, all the offloading schemes operated in a similar way; this was because the workload was minimal and can be processed at the primary fog nodes without using capacity of neighbors. When the number of vehicles was increased, DT achieved the highest throughput, with ~96% compared to other schemes, which accomplished 94.5, 91, and 90% for ST50, JTORA, and WO, respectively.

### 6.4. Impact of Increasing Number of Neighbors on Delay, Throughputs, and Energy with Various Offloading Schemes

The impact of increasing the number of neighbors was carried out to investigate its impact on overall system performance and to find the optimal number of neighbors that are required. From Figure 8, we observe that as the number of neighbors was increased, the delay decreased for both control loops. However, when a certain number of neighbors was reached (e.g., five neighbors), the delay remained almost stable despite adding further neighbors. This means that the optimal number that is required to achieve minimum delay was reached, and no additional neighbors were needed to save energy consumption of the fog paradigm. The reason for the stable pattern was attributed to the workload, as most of the generated tasks were processed.

We note that DT accomplished the least delay for both control loops as the number of neighbors was increased, compared to the other schemes: ST50, JTORA, and WO. With three neighbors, DT decreased delay by 10.80, 13.38, and 15.29% compared to ST50, JTORA, and WO, respectively. When the number of neighbors was five, the DT scheme reduced delay by 55.94, 70.64, and 72.55% in comparison to ST50, JTORA, and WO, respectively.

Increasing the number of neighbors on throughput showed a similar pattern as increasing the number of neighbors to decrease delay. As the number of neighbors increased, the percentage of processed tasks increased, until a certain number of neighbors was achieved (e.g., five neighbors), after which the pattern remained almost stable. The reason behind the stable pattern was due to the workload, as most of the generated tasks were processed. The reason why the percentage of processed tasks did not reach 100% was that this study implemented an online dynamic system, therefore vehicles were still generating tasks until the end of the simulation; 5% of the total generated tasks were not processed because they were newly generated.

In terms of the comparison with other schemes, DT improved throughputs by 0.10% when the number of neighbors was three, 1.16% when the number of neighbors was four, and 1.11% when the number of neighbors was five, six, seven, eight, nine, and ten, compared to ST50 scheme. When the optimal number of five neighboring fog nodes was reached, the DT processed 95.66% of the total generated tasks, while ST50, JTORA, and WO processed 4.55, 91.06, and 90.88%, respectively. The DT scheme improves throughput compared to other stated schemes as the number of neighboring fog nodes was increased.

The impact of increasing the number of neighbors on energy consumption was investigated with various offloading schemes, as shown in Figure 9. When increasing the number of neighbors, the energy consumption in the system was increased because of operating additional fog nodes. Addressing the impact of increasing the number of neighbors helped to find the optimal number of neighboring fog nodes that was necessary to achieve optimum results. When having five neighbors, the difference between the energy consumed with and without DEC was very low; then as we increased the number of neighbors, the difference started to increase. In the no offloading scheme, the impact of utilizing DEC can be observed, i.e., reducing the wastage of energy by 55.72% when the number of neighbors was three, and up to 80.74% when the number of neighbors was ten. This method can also be applied to ST50 and DT, as DEC saved up to 38.58 and 32.16% of energy for each scheme, respectively, when the number of neighbors was ten. When comparing ST50 to DT after applying DEC, more energy was consumed with DT. This was because of the nature of this scheme, as more tasks were processed in DT than ST50, so the energy consumed by processing these tasks caused an increase in overall energy consumption in the system.

## 7. Conclusions

In this paper, we studied the problem of computational offloading and resource management in online fog computing systems and proposed a dynamic offloading threshold that allows a fog node to adjust its threshold dynamically, with a combination of two efficient and effective algorithms: dynamic task scheduling (DTS) and dynamic energy control (DEC). Our proposed scheme exploited the available resources of nearby fog nodes and the remote cloud, selecting the best candidate to handle the overloaded tasks. Moreover, our proposed approach made dynamic decisions as to when to increase/decrease the offloading threshold, which in turn determined whether the incoming task was processed locally at the primary fog node or offload to the best neighbor, based on the states of the fog node’s resources, and its neighbors. Therefore, once the primary fog node was considered congested (e.g., reaching its offloading threshold), it tended to migrate its workloads to the best neighbor.

The performance of the proposed approach was evaluated in terms of average round trip time, throughputs and total energy consumed at fog nodes. In addition to that performance comparisons with more recent offloading schemes were presented to validate the efficiency of the proposed solution. Furthermore, the effect of increasing the number of vehicles was addressed, this was to analyze the performance of the proposed algorithm in cases where traffic congestion occurred in a specific region. Along with that, the impact of the increasing number of neighbors was investigated to examine how the system would perform in situations where there were more available neighbors willing to help. Various numerical results were included, and the performance evaluations were presented to illustrate the effectiveness of the proposed scheme and demonstrate the superior performance over existing schemes.

For future work, one can consider the impact of latency and energy overhead caused by switching on/off fog nodes. Moreover, considering task offloading in an environment that takes user mobility into account. An interesting direction for future research is to examine how latency, energy consumption, security could be optimized simultaneously.

## Figures and Tables

**Figure 1 sensors-21-02512-f001:**
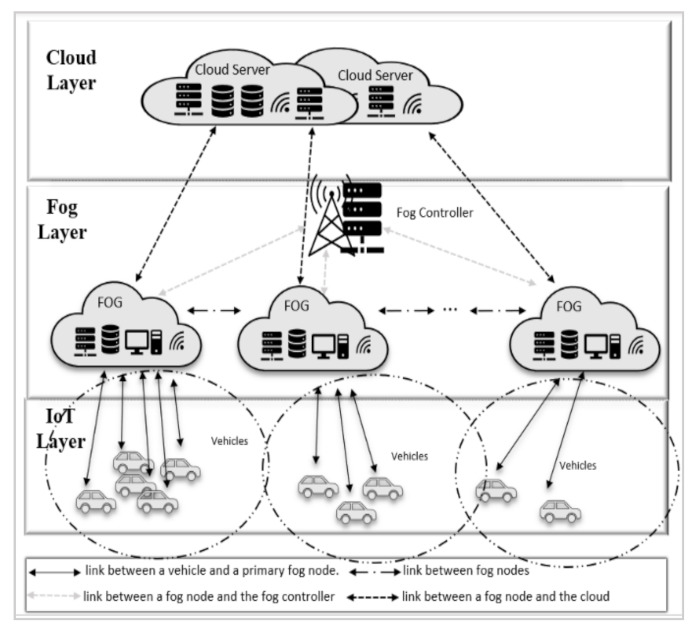
Fog Computing Model.

**Figure 2 sensors-21-02512-f002:**
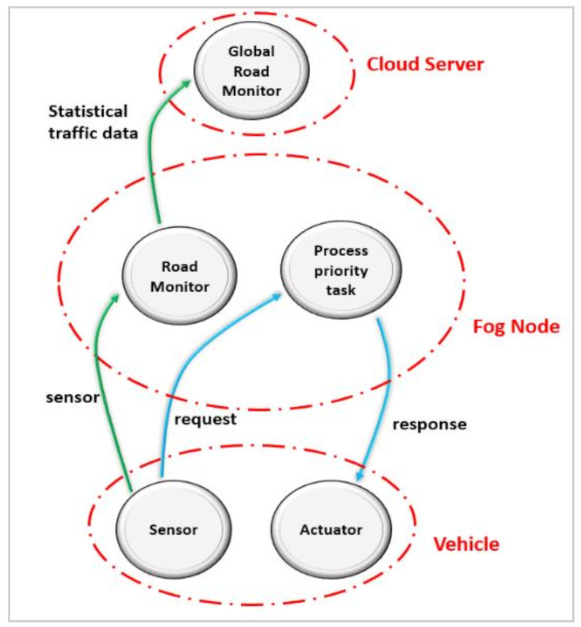
Directed Acyclic Graph (DAG) of the application model.

**Figure 3 sensors-21-02512-f003:**
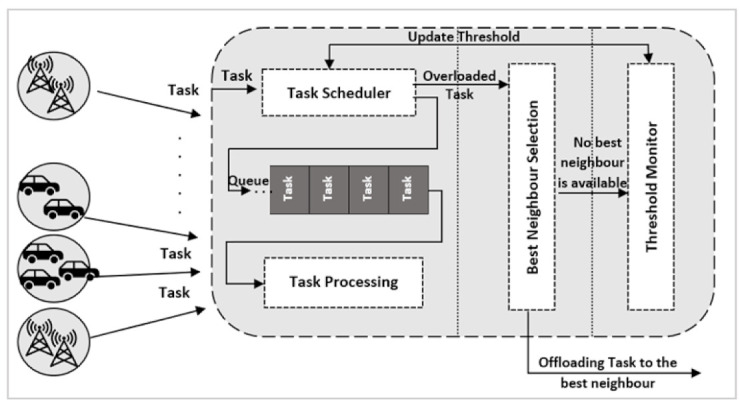
Fog Node Architecture Model.

**Figure 4 sensors-21-02512-f004:**
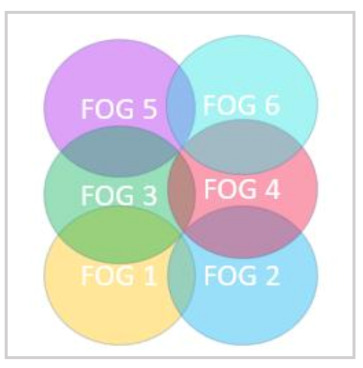
Overlapping Fog Nodes.

**Figure 5 sensors-21-02512-f005:**
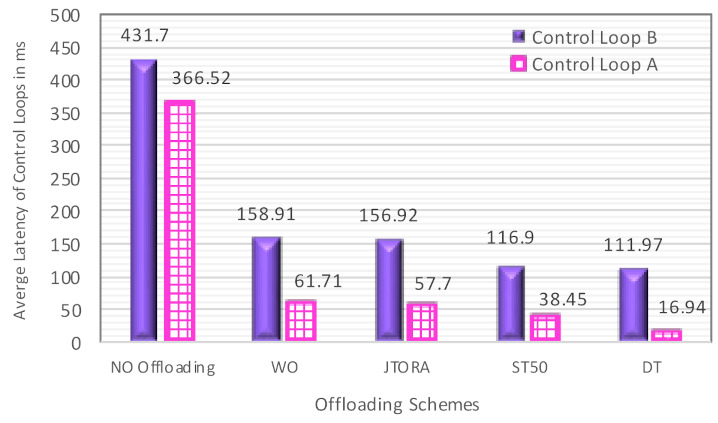
The Comparison of the Average Latency with Various Offloading Schemes.

**Figure 6 sensors-21-02512-f006:**
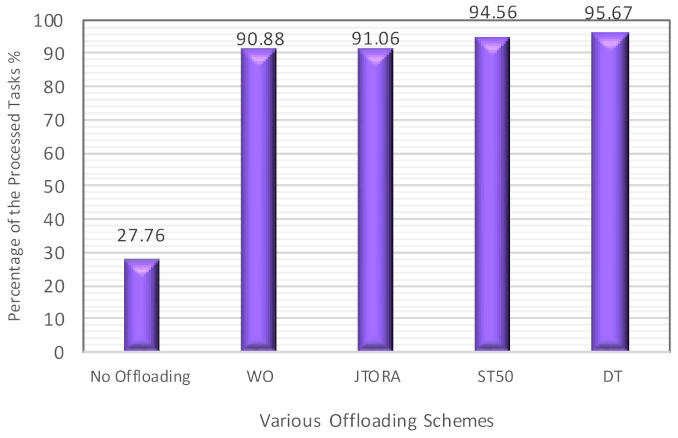
The Comparison of the Throughputs with Various Offloading Schemes.

**Figure 7 sensors-21-02512-f007:**
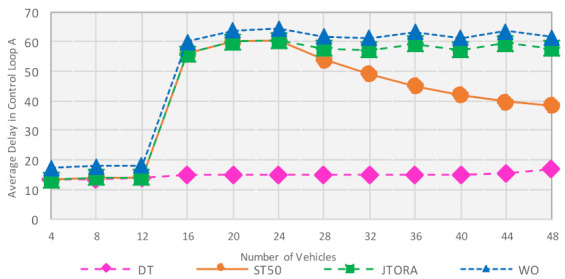
Impact of Increasing Number of Vehicles on Average Delay with Different Offloading Schemes.

**Figure 8 sensors-21-02512-f008:**
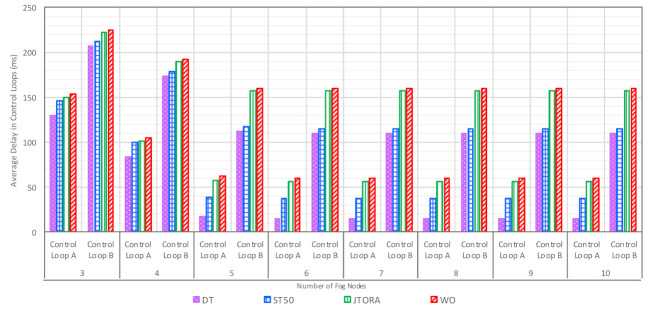
Impact of Increasing Number of Neighbors on Average Delay in Control Loops with Various Offloading Schemes.

**Figure 9 sensors-21-02512-f009:**
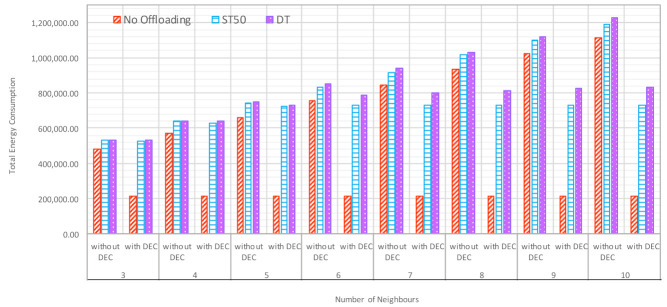
Impact of Increasing Number of Neighbors on Energy Consumption with Various Offloading Schemes.

**Table 1 sensors-21-02512-t001:** Computational Offloading State-Of-Art Comparison.

Ref	Offloading Deployment	Architecture Model				Fog Cooperation	Offloading Threshold	Communication Type		Objectives						Evaluation Tool
		IoT-Fog	IoT-Fog-Cloud	Fog-Cloud	Fog			Vertical	Horizontal	Delay	Energy				Throughput	
											Yes	Relation with Delay	Which Energy	Which Device		
Tang et al. [18]	Offline	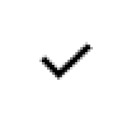	-	-	-	X	static	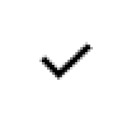	X	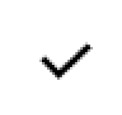	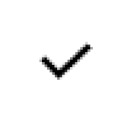	Energy constraint	Energy for data processing and data transmission	IoT	X	MATLAB
Wang and Chen [19]		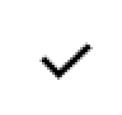	-	-	-	X	static	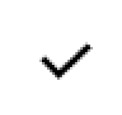	X	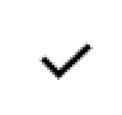	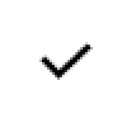	Energy constraint	Energy for local processing, processing at fog nodes, transmitting tasks	IoT devices and Fog nodes	X	Simulation
Liu et al. [11]		-	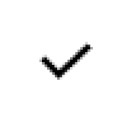	-	-	X	static	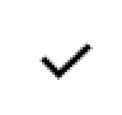	X	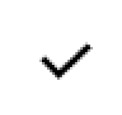	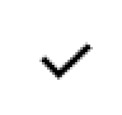	Trade-off	Energy spent by local processing and transmitting tasks	Mobile devices	X	Simulation
Mukherjee et al. [20]		-	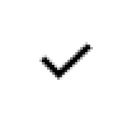	-	-	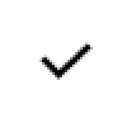	static	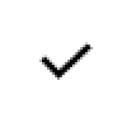	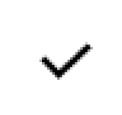	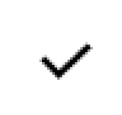	X	X	X	X	X	Monte Carlo simulations
Zhu et al. [13]		-	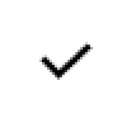	-	-	X	static	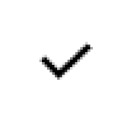	X	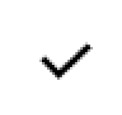	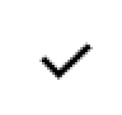	Offloading policy is designed to minimize task execution delay and to save energy	The energy spent by uploading and receiving tasks	Mobile devices	X	Simulation (MATLAB)
Mukherjee et al. [21]		-	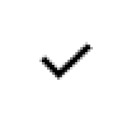	-	-	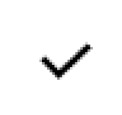	static	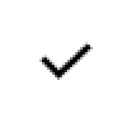	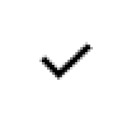	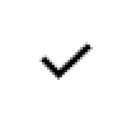	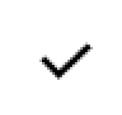	Minimize total systems cost which includes total energy consumption & total processing delay	Energy consumed by local computing and uploading tasks to fog nodes	End user devices	X	MATLAB
Chen and Hao [14]		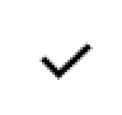	-	-	-	X	static	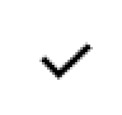	X	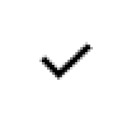	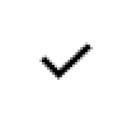	Battery capacity	Energy for transmitting tasks to edge devices and for local processing	End user devices	X	Simulation
Sun et al. [22]		-	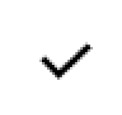	-	-	X	static	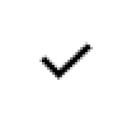	X	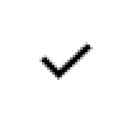	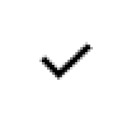	Selects the lowest overhead cost which involves total computational time plus total energy spent by either processing task at IoT devices or transmission tasks to fog or cloud	Processing and transmission power	IoT, fog nodes, cloud servers	X	iFogSim
Zhao et al. [12]		-	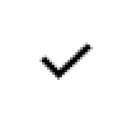	-	-	X	static	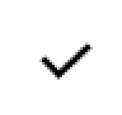	X	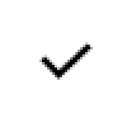	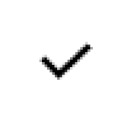	Minimizes the system cost which is the total offloading latency and the total energy consumption	Transmission and processing energy	Whole System: end user devices, fog nodes, and cloud servers	X	Simulation
Meng et al. [23]		-	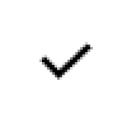	-	-	X	static	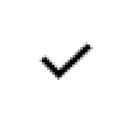	X	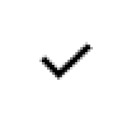	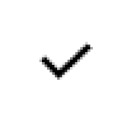	Minimizing energy given delay constraint	Transmission + computational energy	Mobile terminal, fog nodes, and cloud servers	X	Simulation
Xiao and Krunz [8]		-	-	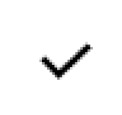	-	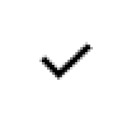	static	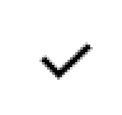	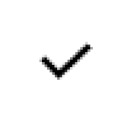	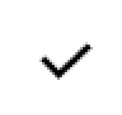	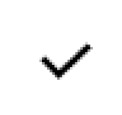	Trade-off	Transmission energy	Fog nodes	X	Simulation
Yousefpour et al. [16]	Online	-	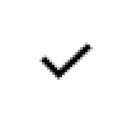	-	-	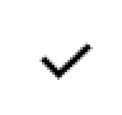	static	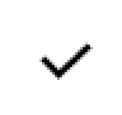	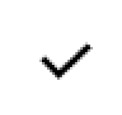	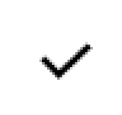	X	X	X	X	X	Simulation
Yin et al. [24]		-	-	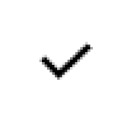	-	X	static	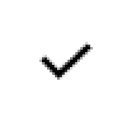	X	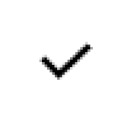	X	X	X	X	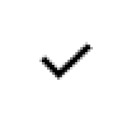	Simulation
Al-Khafajiy et al. [15]		-	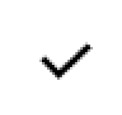	-	-	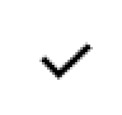	static	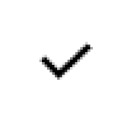	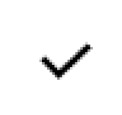	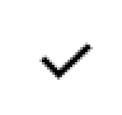	X	X	X	X	X	MATLAB-based simulation
Gao et al. [17]		-	-	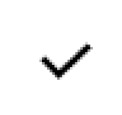	-	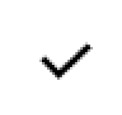	static	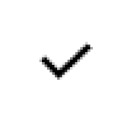	X	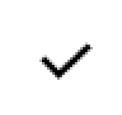	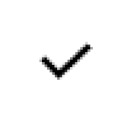	Trade-off	Processing and transmitting tasks between fog nodes	Fog nodes	X	Simulation
Mukherjee et al. [25]		-	-	-	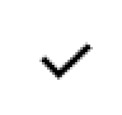	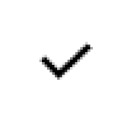	static	X	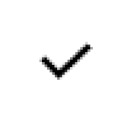	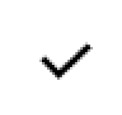	X	X	X	X	X	Simulation
Alenizi and Rana [1]		-	-	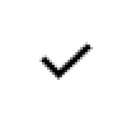	-	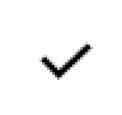	static	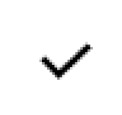	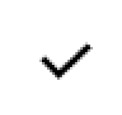	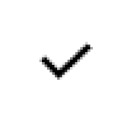	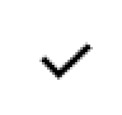	X	POWERING ON and processing tasks	Fog nodes	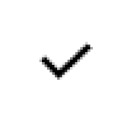	iFogSim
The proposed approach		-	-	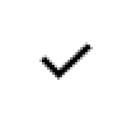	-	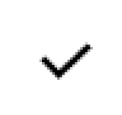	Dynamic	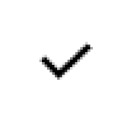	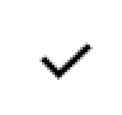	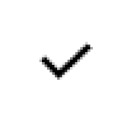	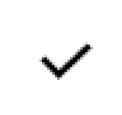	X	POWERING ON and processing tasks	Fog nodes	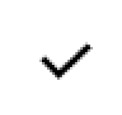	iFogSim

**Table 2 sensors-21-02512-t002:** Example of Offloading Threshold set for Fog Node A and its neighbors.

Fog Node Type	Primary Fog Node	Neighboring Fog Nodes			
	Fog node A	Fog node B	Fog node C	Fog node D	Fog node E
Threshold	9 ms	9 ms	9 ms	9 ms	9 ms

**Table 3 sensors-21-02512-t003:** Example of Offloading Threshold set for Fog Node B and its neighbors.

Fog Node Type	Primary Fog Node	Neighboring Fog Nodes			
	Fog node B	Fog node A	Fog node F	Fog node G	Fog node H
Threshold	6 ms	6 ms	6 ms	6 ms	6 ms

**Table 4 sensors-21-02512-t004:** Description of Parameters used for Dynamic Threshold Algorithm.

Symbol	Description
$\delta^{n}$	Refers to the initial offloading threshold and the current threshold.
VQ	Average queueing delay of all the neighbors
$\delta^{n+1}$	New offloading threshold.
x	$x=\delta^{n}/2$.
Ns	All neighboring fog nodes
Qs	Set of all queueing delay of all its neighbors
QNeighbours	Set of all neighbors and their queueing delay
α	When the queuing delay reaches this threshold, the fog node might consider decreasing its offloading threshold.
N	The best neighbor fog node
T	The arrival task
Q	Queuing delay in the primary fog node
p	A number bigger than zero that determines how much to modify the offloading threshold based on the current offloading threshold
Q_N_	Queueing delay of one neighbor

## Data Availability

Data sharing not applicable.

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
