# Peer review of "Dynamically Controlling Offloading Thresholds in Fog Systems"

_sensors, 2021, doi:10.3390/s21072512_

Round 1

Reviewer 1 Report

This paper propose a fog computing framework to perform dynamic resource management and provide computational offload for IoT devices. This paper extends the authors' prior work by proposing a dynamic offloading threshold based on service delay. Sufficient state-of-the-art literature has been surveyed. 

Some places to be further improved:

  1. The authors model the offloading problem as an optimization problem, but do not provide details regarding the algorithm to solve the problem. Furthermore, what is the computational complexity of the offloading algorithm itself?
  2. It is not clear whether the offloaded computation will be sent to the cloud end or not when congestion occur. If yes, how does the decision process work? For example, during rush hours, there may be much more vehicles waiting for local services, causing congestions within a certain area. 
  3. For the experiment part, the authors involve iFogsim for experiment evaluation. How are the IoT computation workload simulated?  
  4. Page 17, there is an incomplete sentence in line 6 ("Throughput" part). Please make sure to proof read the paper. 

Reviewer 2 Report

The submission presents an original concept of the Authors, supported by extensive simulation experiments. The presentation quality (including the lauoyt of the paper) is very good, however the 'Conclusions' section could be extended to summarize interesting results obtained by the Authors.

Some minor errors should be corrected before acceptation of the final version:

  -- page #2 - remove double dots in the last line;

  -- page #6-7 - Table 1 contains good summary of the state-of-art, however is rather difficult for analysis; also its editorial form should be cheched again (e.g. word wrapping in the last row);

  -- page #8 - spell checking in the last two paragraphs is required;

  -- page #11 (just below "Communication constraints") - text justification should be improved;

  -- page #11, eq. (2) - squares must be added in the formula for distance computation;

  -- page #11 (the line above "Processing Constraints") - the value of 'c' constant (light speed) is wrong, it is also a nominated quantity;

  -- page #11, eq. (4) the names inside the summation symbol look slightly strange, the second sum does not have limits;

  -- page #13 (the line above "3.2.3 Between Fog Nodes and the Cloud") - the same problem with the light speed constant;

  -- page #22-24 - the format of all citations should be uniform.

Reviewer 3 Report

The authors of this paper have presented the use of fog computing to dynamically control the offloading network threshold. Moreover, they also looked at the management side of fog resources in some dynamic systems. The paper has the following issues:

  1. The contributions are not novel. There many papers that have discussed this topic!
  2. The system validation section is weak and needs more results and discussions!
  3. The figures are excessively large!
  4.  Algorithm 1 Dynamic Offloading Threshold is basic and not linked to the system model.
  5. What is the point from Fig. 5?
  6. The sensors in the paper are not clearly shown or discussed! This is a sensor-based journal.
  7. Other technologies such as sdn can be discussed. I recommend you to look after the following: An SDN architecture for time sensitive industrial IoT, Reinforcing Cloud Environments via Index Policy for Bursty Workloads.

Round 2

Reviewer 3 Report

The authors have been given a chance to articulate their answers to the comments, however, their response is shallow! They did not give it a proper time and effort, therefore, and because of the pointed issues, I cannot accept it in this round!

Author Response

thank you for your feedback. the manuscript has been updated